# Pre-Hospital Management of Critically Ill Patients with SARS-CoV-2 Infection: A Retrospective Multicenter Study

**DOI:** 10.3390/jcm9113744

**Published:** 2020-11-21

**Authors:** Pierrick Le Borgne, Mathieu Oberlin, Adrien Bassand, Laure Abensur Vuillaume, Yannick Gottwalles, Marc Noizet, Stéphane Gennai, Florent Baicry, Deborah Jaeger, Nicolas Girerd, François Lefebvre, Pascal Bilbault, Tahar Chouihed

**Affiliations:** 1Emergency Department, University Hospital of Strasbourg, 1 Avenue Molière, 67000 Strasbourg, France; mathieu.oberlin@chru-strasbourg.fr (M.O.); florent.baicry@chru-strasbourg.fr (F.B.); pascal.bilbault@chru-strasbourg.fr (P.B.); 2INSERM (French National Institute of Health and Medical Research), UMR 1260, Regenerative NanoMedicine (RNM), Fédération de Médecine Translationnelle (FMTS), University of Strasbourg, 67000 Strasbourg, France; 3Emergency Department, University Hospital of Nancy, 29 Avenue du Maréchal de Lattre de Tassigny, 54035 Nancy, France; a.bassand@chru-nancy.fr (A.B.); d.jaeger@chru-nancy.fr (D.J.); t.chouihed@chru-nancy.fr (T.C.); 4Emergency Department, Regional Hospital of Metz-Thionville, 1, Allée du Château, 57530 Ars-Laquenexy, France; l.abensurvuillaume@chr-metz-thionville.fr; 5Emergency Department, General Hospital of Colmar, 39 Avenue de la Liberté, 68024 Colmar, France; yannick.gottwalles@ch-colmar.fr; 6Emergency Department, Mulhouse hospital, 20 Avenue du Dr René Laennec, 68100 Mulhouse, France; marc.noizet@ghrmsa.fr; 7Emergency Department, Reims University Hospital, 45 Rue Cognac-Jay, 51100 Reims, France; sgennai@chu-reims.fr; 8Centre d’Investigations Cliniques-1433, and INSERM U1116, F-CRIN INI-CRCT, Université de Lorraine, 54000 Nancy, France; n.girerd@chru-nancy.fr; 9Department of Public Health, University Hospital of Strasbourg, 1 Place de l’Hôpital, 67000 Strasbourg, France; francois.lefebvre@chru-strasbourg.fr

**Keywords:** pre-hospital care, acute respiratory distress syndrome, intensive care unit, COVID-19, critical care, phenotypes

## Abstract

Introduction: The COVID-19 outbreak had a major impact on healthcare systems worldwide. Our study aims to describe the characteristics and therapeutic emergency mobile service (EMS) management of patients with vital distress due to COVID-19, their in-hospital care pathway and their in-hospital outcome. Methods: This retrospective and multicentric study was conducted in the six main centers of the French Greater East region, an area heavily impacted by the pandemic. All patients requiring EMS dispatch and who were admitted straight to the intensive care unit (ICU) were included. Clinical data from their pre-hospital and hospital management were retrieved. Results: We included a total of 103 patients (78.6% male, median age 68). In the initial stage, patients were in a critical condition (median oxygen saturation was 72% (60–80%)). In the field, 77.7% (CI 95%: 71.8–88.3%) were intubated. Almost half of our population (45.6%, CI 95%: 37.1–56.9%) had clinical Phenotype 1 (silent hypoxemia), while the remaining half presented Phenotype 2 (acute respiratory failure). In the ICU, a great number had ARDS (77.7%, CI 95% 71.8–88.3% with a PaO_2_/FiO_2_ < 200). In-hospital mortality was 33% (CI 95%: 24.6–43.3%). The two phenotypes showed clinical and radiological differences (respiratory rate, OR = 0.98, *p* = 0.02; CT scan lesion extension >50%, OR = 0.76, *p* < 0.03). However, no difference was found in terms of overall in-hospital mortality (OR = 1.07, *p* = 0.74). Conclusion: The clinical phenotypes appear to be very distinguishable in the pre-hospital field, yet no difference was found in terms of mortality. This leads us to recommend an identical management in the initial phase, despite the two distinct presentations.

## 1. Introduction

As of late 2019, the medical and scientific worlds have been confronting a pandemic caused by a novel coronavirus called severe acute respiratory syndrome coronavirus 2 (SARS-CoV-2) [1]. The clinical course of SARS-CoV-2 infection ranges from an asymptomatic presentation to acute, sometimes hypoxemia-inducing, viral pneumonia. In a minority of cases (nearly 5%), the novel virus leads to acute respiratory distress syndrome (ARDS) requiring intensive care unit admission and prolonged mechanical ventilation [2]. SARS-CoV-2 infection severity and prognosis appear to be related to both respiratory impact of the disease and patients’ characteristics, including underlying comorbidities and age [3]. These critical patients seem to display the classic criteria of ARDS (severe hypoxemia and acute onset of bilateral infiltrates) [2]. However, some of these ARDS forms seem to be atypical; even though ARDS presentation is already occasionally heterogeneous, it seems to be even more so in COVID-19 infection [4].

Several recent studies have raised the hypothesis of several phenotypes in critical SARS-CoV-2 infection [5,6]. *Gattinoni et al.* [7] described two different phenotypes (L and H) and recommended a distinct therapeutic approach in the management of severe patients with SARS-CoV-2. They postulated that patients with L-type phenotype appear to present atypical forms of ARDS while those with H-type phenotype seem to develop a more standard form of ARDS. This hypothesis can be elaborated in dual ways: either as two phenotypes with distinct manifestations of severe SARS-CoV-2 infection potentially requiring different management strategies, or as a transition, a continuous spectrum of the disease where type L progresses towards type H [8]. Moreover, field observations along with recent publications discern a peculiar presentation with gravely hypoxemic patients showing no sign of clinical respiratory distress (“silent or happy” hypoxemia) while others exhibit a more conventional presentation of acute respiratory failure and hypoxemia [9].

During the outbreak, emergency calls increased four to five-fold, which increased the risk of overwhelming emergency departments (EDs). However, there were some healthcare resources devoted to the pre-hospital management of patients in critical distress [10]. In this strategy, the most critical patients were managed by an emergency mobile service (EMS) and directly admitted to intensive care units (ICUs), bypassing emergency departments and avoiding overcrowding.

The aim of our study was to describe the clinical characteristics, the pre-hospital management and the outcome of these early managed acute patients in the Greater East region of France.

## 2. Methods

### 2.1. Settings

This retrospective, multicentric study was conducted in the six main emergency departments in the Greater East region of France (Strasbourg, Nancy, Reims, Colmar, Mulhouse and Metz-Thionville). This area was one of the most impacted in Europe by the pandemic with nearly 3400 deaths and more than 10,000 infected patients as of the end of May.

### 2.2. Study Population

From 1 March to 20 April, during the pandemic period in France, we collected and included data of all SARS-CoV-2-infected patients managed by EMS according to dispatching criteria (respiratory distress, de-saturation, consciousness impairment or shock) and who were admitted straight to ICUs. In addition, we included all patients admitted to the ED without necessarily EMS, but who were secondarily admitted within two hours to the ICU. Based on clinical experience during the first weeks of the pandemic, we have defined two clinical phenotypes in our study population: Type 1 corresponding to silent hypoxemia without acute respiratory distress and Type 2 associating hypoxemia and clinical respiratory failure (polypnoea, pulling, paradoxical breathing, use of accessory muscles).

The diagnosis of SARS-CoV-2 was made by nasopharyngeal swab Rt-PCR, and all patients included in this study had at least one positive test during their hospital stay. Patients who presented directly to an emergency department were excluded.

### 2.3. Data Collection

For each patient, we collected epidemiological and demographic data such as age, sex and medical history, along with the motive of their emergency call. Clinical data were also collected including vital signs, main diagnosis, pre-hospital management, the need for mechanical ventilation and duration of on-site first aid care. Then, we studied the overall hospital length of stay: firstly in the ICU, including need for invasive support care (mechanical ventilation, dialysis, etc.) and the occurrence of complications (pulmonary embolism), then secondly in the post-ICU departments. Severity of illness was assessed using the Simplified Acute Physiology Score II (SAPS II) [11]. Finally, we collected the ICU and overall hospital stay mortality.

### 2.4. Ethics

This study was approved by Strasbourg University’s ethics review board (reference: CE-2020-64), which, in accordance with French legislation, waived the need for informed consent from patients whose data were entirely retrospectively studied [12].

### 2.5. Statistical Analysis

The descriptive statistical analysis of the quantitative variables was performed by providing the frequencies, cumulative frequencies, proportions and cumulative proportions of each value. Whenever useful, cross-tabulations were given with frequencies, proportions by row, proportions by column, and proportions of the total for each box in the table. For each quantitative variable, the location parameters (mean, median, minimum, maximum, first and third quartiles) and the dispersion parameters (variance, standard deviation, range, interquartile range) were given. The quantitative variables were described with histograms (with different bin width and with kernel density estimation). Normality of the distributions was tested using a normality test, such as the Shapiro–Wilk test, and was assessed graphically using a normal quantile plot. To compare the continuous covariates, Student tests or Wilcoxon tests in the case of non-normality were performed. To compare the categorical covariates, Chi-Squared tests or Fisher tests were performed. Then, a multivariate logistic model was performed with the statistically significant and clinically relevant covariates. A backward stepwise method was used based on Akaike Information Criterion (AIC). We also performed subgroup analyses comparing these data and the overall survival rates, and the survival rates between in-hospital survivors and non-survivors and between the two clinical phenotypes (hypoxemia with, versus without, acute respiratory failure). Analyses were performed with R software (version 4.0.2) in its most up-to-date version at the time of the analysis, as well as with all the software packages required to carry out the analysis.

## 3. Results

### 3.1. Clinical Characteristics

From 1 March to 20 April, we included 103 patients. The median age was 68 years (60–74). The majority of patients were male (78.6%, CI 95%: 72.8–89.1%), and most of them were obese (36.9%, CI 95%: 27.6–47%) or overweight (27.2% CI 95%: 18.9–36.8%). One third of them had a medical history of diabetes (33%, CI 95%: 24.1–43%), and 59.2% (CI 95%: 49–68.9%) were being treated for hypertension. Approximately one third of all patients had a cardiovascular (35%, CI 95%: 26.5–45.4%) or a respiratory (35%, CI 95%: 26.5–45.4%) medical history (Table 1).

### 3.2. Pre-Hospital Management

Upon arrival of EMS, the majority of patients were in critical condition: respiratory rate was 30/min (26–40), and oxygen saturation was 72% (60–80) without oxygen. First, all patients required high-flow oxygen therapy (15 L/min). Half of the patients receiving pre-hospital care showed clinical signs of respiratory failure (53.4%, CI 95%: 43.3–63.3%), and nearly a quarter of them were restless or confused (24.3%, CI 95%: 16.4–33.7%).

From the onset of symptoms to the emergency call, patients reported a median duration of 7 days (4–10 days). The median duration lasted 1 day (1–2 days) from the onset of acute respiratory symptoms to dispatch center. A majority of patients 77.7% (CI 95%: 71.8–88.3%) received endotracheal intubation in the field, before ICU admission. Thus, they were transported to the ICU on mechanical ventilation (median FiO_2_: 80%, 60–80%), while all others were transported on spontaneous ventilation (15 L/min of high-flow oxygen therapy).

### 3.3. Initial Management

The first arterial blood gas in the ICU showed a median pH of 7.40 (7.28–7.47), a median PaO_2_ of 73 mmHg (60–102) and a median PaCO_2_ of 37.5 mmHg (31.3–44). Median arterial lactate was 1.6 mmol/L (1.1–2.2) (Table 1). Shortly after their ICU admission, a total of 69 patients (67%, CI 95%: 59.7–78.4%) underwent a chest CT scan to assess pulmonary involvement. Among the radiological findings, lung abnormalities were typical of COVID-19 infection in 40 patients (58%, CI 95%: 46.4–69.6%) and compatible in 23 others (33.3%). The extent of lung damage was greater than 50% for 61.3% (CI 95%: 34.4–50.2%) of patients.

### 3.4. ICU Stay

Only a few patients (11.7%) were intubated upon arrival to the ICU, and one patient died very early after admission. The first 24 h assessment, using the PaO_2_/FiO_2_ ratio, found that the majority of patients (91.3%, CI 95%: 88.4–99.6%) had ARDS: 41.8% of patients had severe ARDS (P/F ratio < 100), 35.9% of patients had moderate ARDS (P/F ratio: 100–200), and 13.6% of patients had mild ARDS (P/F ratio: 200–300). In our study population (n = 103), the median duration of mechanical ventilation was 12 days (4–23 days), and the median SAPS II score was 51 (40–62). Over half of the patients received prone position sessions (61.2%), only a few patients (8.7%) benefited from veno-venous extracorporeal membrane oxygenation (ECMO), and almost a quarter of the patients required dialysis (22.3%, CI 95%: 14.7–31.2%)). During their stay in the ICU, 10 patients (9.7%, CI 95%: 4.1–15.9%) were diagnosed with pulmonary embolism. Moreover, eight patients (7.8%, CI 95%: 2.6–13%) were transferred abroad (to bordering European countries), and two patients (1.9%, CI 95%: 0–4.5%) were lost to follow-up.

### 3.5. Outcome

Mortality in the ICU was 31.1% (CI 95%: 22.2–40%), and intra-hospital mortality was 33% (CI 95%: 24.6–43.3%). The median stay in the ICU lasted 16 days (5–29 days) while the median overall hospital length of stay (ICU and post-ICU wards) lasted 25 days (11–39 days). We performed a comparison between the surviving and non-surviving subgroups. Patients who did not survive were significantly older (72 vs. 65 years, *p* = 0.02), had more severe pre-hospital clinical presentation (higher oxygen need), higher creatinine levels on admission (*p* = 0.02) and higher incidence of acute pulmonary embolism (*p* = 0.03) (Table 1).

### 3.6. Clinical Phenotypes (Type 1: Silent Hypoxemia versus Type 2: Hypoxemia with Clinical Acute Respiratory Failure)

In our population, Type 2 patients are more typically in a state of respiratory failure (clinical signs of respiratory distress, higher respiratory rate, more neurologically impaired) than Type 1 patients and are, *de facto*, more often intubated in a pre-hospital setting (71.1% versus 98.2%, *p* < 0.01). These patients (Type 2) have more extensive respiratory lesions on chest CT scans (>50% extension, *p* < 0.01). SAPS II score is higher in Type 2 patients (43.5 vs. 53, *p* = 0.04). The length of stay in the ICU is similar in the two subgroups (*p* = 0.49), as are the severity of ARDS (*p* = 0.81) and the duration of mechanical ventilation (*p* = 0.55). ICU mortality (31.9% versus 30.4%, *p* = 0.86) and in-hospital mortality are, as well, identical in the two clinical phenotypes (Table 2).

### 3.7. Multivariate Analysis Type 1 versus Type 2 Phenotypes

When comparing adjusted results, demographic characteristics and medical history did not differ except for history of cardiovascular diseases (OR = 0.73, CI 95%: 0.55–0.96, *p* = 0.03), which was more frequent in Phenotype 2. No difference was found in the pre-hospital management and clinical presentation except for respiratory rate (OR = 0.98, CI 95%: 0.96–1, *p* = 0.02) and in-field intubation (OR = 0.67, CI 95%: 0.51–0.88, *p* < 0.01). Regarding the ICU stay, the duration of mechanical ventilation was longer in the case of Phenotype 1 (OR = 1.03, CI 95%: 1.01–1.06, *p* = 0.02). The total length of hospital stay was also prolonged for Phenotype 1 (OR = 1.02, *p* = 0.04). However, concerning ICU and in-hospital mortality (OR = 1.07, CI 95%: 0.72–1.59, *p* = 0.75), we found no differences according to clinical phenotype (Table 3 and Figure 1).

## 4. Discussion

In a population of severe patients suspected of SARS-CoV-2 infection and managed in a pre-hospital setting, our work highlights the value of early assessment and admission of silent hypoxemia patients to a high dependency unit and acute respiratory failure patients straight to the intensive care unit. To the best of our knowledge, this is the first study to assess critical COVID-19 patients in the pre-hospital setting.

### 4.1. Study Population

We are all currently facing unexplored frontiers with this unknown virus, whose physiopathology is only partially identified. The clinical characteristics of the patients in our study are fairly standard and consistent with those of recent publications: the patients were mostly elderly men with a medical history of hypertension [13,14,15]. Concerning ICU stay, the large majority of these patients had severe ARDS, required prolonged mechanical ventilation and stayed in hospital for several weeks. In-hospital mortality was relatively high at 33% [16,17].

Our study identified two important delays. First, we found a delay in clinical worsening comparable to that in the literature: disease progression and clinical worsening occurred on around day 7 of infection [18]. On the other hand, we found that the length of pre-hospital care was particularly short and lasted less than one hour, thus limiting the loss of chance for patients [19].

During this global crisis, one of the main concerns of medical organizations around the world was not to overwhelm the intensive care, regular hospitalization and emergency medicine capacities that were available. This involved the theoretical goal of providing a ventilator for every patient when necessary (triage and rationing) [20]. At the peak of the outbreak, when intensive care units and EDs reached their maximum capacity, it was hypothesized that several phenotypes existed in the critical SARS-CoV-2 infection. This new concept emerged as a potential solution to stratify the severity of patients with COVID-19 [21]. Thus, it would be interesting to be able to stratify critical COVID-19 patients as soon as possible in order to send them either to a high dependency unit or to an intensive care unit.

### 4.2. Beyond Type 1 and 2 Clinical Phenotypes

We have observed that patients initially tolerate severe hypoxemia well, then the situation worsens rapidly in a span of a few minutes to the point of requiring urgent intubation. This may explain the hypothesis of *Gattinoni et al.* [7], among others, that both phenotypes exist in critical SARS-CoV-2 infection. Moreover, we also observe these two phenotypes in our population, but it is essential to understand that Type 1 patients (silent or happy hypoxemia) could be wrongly reassuring because they could present a swift and brutal clinical deterioration warranting their early admission to ICUs [22,23]. Indeed, and this already at the Type 1 step, there is evidence that ventilation enhancement is beneficial to the patient. For example, prone decubitus therapy would improve both the oxygenation rate and prognosis of mechanically ventilated patients, but also helps patients under spontaneous ventilation by significantly reducing the rate of intubation and improving mortality [23,24]. Several explanations have been developed to highlight these clinical disparities. Some authors have suggested the neuro-invasive potential of SARS-CoV-2, while others have advanced the affinity of viral proteins for hemoglobin inducing decreased oxygen transport [25,26,27]. A few have compared this presentation of hypoxemia to that which occurs during acute exposure at high altitude [28]. There may also have been different phenotypes related to the sequence diversity in SARS-CoV-2 proteins associated with viral pathogenicity and transmission in Europe (several countries could be associated with a certain, or even several, virus clades and mutations; frequency of COVID-19 symptoms also varies between these countries) [29]. In addition, one of the emerging therapeutic approaches could be guided by other studies including patient clinical data (sequence polymorphisms and symptoms), thus suggesting that sequence determination of the associated infectious genome should be combined with clinical data to give a better understanding of SARS-CoV-2 pathogenicity [30]. Finally, the theory associating vascular dysfunction (coagulopathy), immuno-thrombosis and deregulated inflammation could also be of real interest in trying to explain these phenotypic variations [31].

This qualification and gradation in terms of patient severity was only possible with the presence of a medically led mobile team on site. Our results suggest that these severe patients should be treated identically in the early stages of the disease. However, since mortality is comparable in the two clinical phenotypes, further studies with a larger population of patients are required. Moreover, it seems plausible at this stage of our research to consider these two phenotypes rather as a continuous spectrum of the same disease than two distinct entities to be managed differently [21,32].

### 4.3. Limitations

Our study manifests some limitations. Firstly, it should be noted that our study was conducted retrospectively. As a matter of fact, it was difficult to build this pre-hospital cohort prospectively given the workload and pressure submerging various ICUs and EDs during the pandemic. Secondly, our small population did not allow us to estimate, with sufficient precision, the mortality risk factors of these patients with severe SARS-CoV-2 infection. However, it should be taken into account that our findings are consistent with those of recent publications. Finally, the Greater East region of France, where this study was performed, is probably the only area capable of sharing this type of pre-hospital data on critical SARS-CoV-2 patients, as other French regions were spared the magnitude of the pandemic (outside the Paris area).

## 5. Take-Home Messages

When a suspected COVID-19 patient is managed in a pre-hospital setting during the key period between day 5 and day 7, it is important to discuss an early admission to ICU or a careful monitoring in a high dependency unit, where patients can benefit from alternative ventilation methods that can avoid intubation and/or to continue critical care management initiated in the pre-hospital setting.

## 6. Conclusions

Severe SARS-CoV-2 infections have, undeniably, unveiled to physicians two distinct clinical presentations. Nonetheless, the confounding yet peculiar first phenotype (“happy” hypoxemia) should be managed similarly to the second phenotype (hypoxemia with clinical acute respiratory failure), which includes an early admission to the ICU or close supervision in a high dependency unit for appropriate vital support.

## Figures and Tables

**Figure 1 jcm-09-03744-f001:**
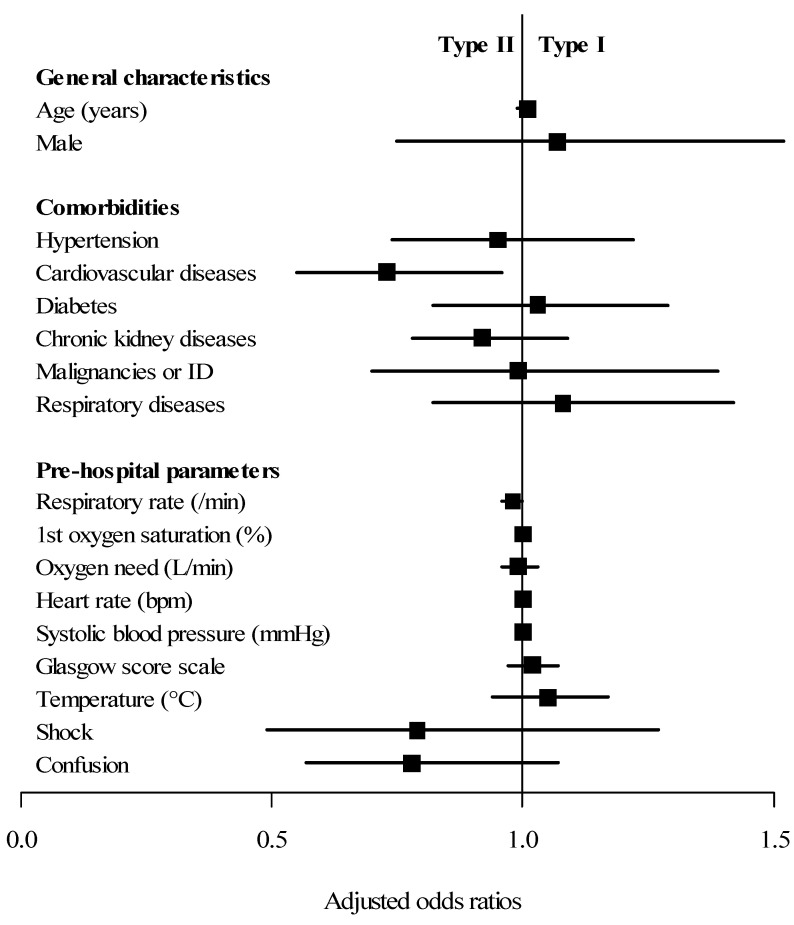
Multivariate analysis of factors associated with the difference between Type 1 and Type 2 clinical profiles. Type 1 is defined in our study as “silent/happy” hypoxemia without symptoms of ARF. Type 2 corresponds to patients who have both hypoxemia and clinical ARF. Abbreviations: ID = immunodeficiency, min = minute, L = liter, ARF = acute respiratory failure (clinical), EMS = emergency medical services (medicalized), CRP = C-reactive protein, CT = computed tomography, ICU = intensive care unit, LOS = length of stay, ARDS = acute respiratory distress syndrome, ECMO = extracorporeal membrane oxygenation, SAPS II = Simplified Acute Physiology Score II.

**Table 1 jcm-09-03744-t001:** Clinical characteristics at baseline and outcome.

General Characteristics	All Patients N = 103	Survivors N = 69	Non-Survivors N = 34	*p*-Value
Age (years)	68 (60–74)	65 (57–73)	72 (66–75)	0.02 *
Male	81 (78.6)	53 (76.8)	28 (82.4)	0.52
**Comorbidities**				
Hypertension	61 (59.2)	43 (62.3)	18 (52.9)	0.36
Cardiovascular diseases	36 (34.9)	22 (31.9)	14 (41.2)	0.35
Diabetes	34 (33.0)	23 (33.3)	11 (32.4)	0.92
Chronic kidney diseases	16 (15.5)	8 (11.6)	8 (23.5)	0.12
Malignencied or ID	12 (11.7)	7 (10.1)	5 (14.7)	0.70
Respiratory diseases	36 (34.9)	25 (36.2)	11 (32.6)	0.70
**Pre-hospital parameters**				
Respiratory rate (/min)	30.0 (26–40)	30.0 (25–38)	34.5 (30–40)	0.18
1st oxygen saturation (%)	72 (60–80)	74 (60–80)	68 (55–80)	0.23
Oxygen need (L/min)	15 (15–15)	15 (12–15)	15 (15–15)	0.03 *
Heart rate (bpm)	103 (85–118)	109 (90–120)	94 (80–110)	0.06
Systolic blood pressure (mmHg)	120 (110–137)	120 (110–140)	122 (110–130)	0.74
Glasgow score scale	15 (14–15)	15 (14–15)	15 (14–15)	0.88
Temperature (°C)	38.0 (37.3–38.8)	38.0 (37.2–38.5)	38.1 (37.3–39.0)	0.31
**Timing**				
Since 1st symptom (days)	7 (4–10)	7 (5–9)	7 (3.3–10)	0.78
Since ARF (days)	1 (1–2)	1 (1–2)	1 (0.3–2)	0.72
With EMS team (min)	47 (30–72)	40 (25–72)	54 (33–74)	0.12
**Pre-hospital care**				
Intubation in the field	80 (87.0)	54 (88.5)	26 (83.9)	1.00
Type 1: Silent hypoxemia	47 (45.6)	32 (46.4)	15 (44.1)	0.83
Type 2: Hypoxemia + ARF	56 (54.4)	37 (53.6)	19 (55.9)	0.83
**Laboratory findings**				
Creatinine (µmol/L)	97 (75–142)	93 (75–122)	109 (89–223)	0.02 *
Lymphocytes (/µL)	675 (470–1053)	680 (470–1060)	630 (470–1030)	0.52
CRP (mg/L)	144 (80–224.3)	142 (78–226)	155 (88–215)	0.84
pH	7.40 (7.28–7.47)	7.41 (7.29–7.47)	7.36 (7.27–7.44)	0.12
PaO2 (mmHg)	73.0 (60.1–102.3)	73.0 (61.0–104.0)	75.0 (55.0–100.0)	0.98
PaCO2 (mmHg)	37.5 (31.3–44.0)	38.0 (33.0–45.0)	37.0 (30.0–42.0)	0.27
Lactate (mmol/L)	1.6 (1.1–2.2)	1.6 (1.1–2.1)	1.8 (1.2–2.8)	0.25
**Radiology findings**				
Typical CT-scan	40 (58.0)	27 (58.7)	13 (56.5)	0.86
Extension >50%	38 (61.3)	27 (65.9)	11 (52.4)	0.30
**ICU stay and outcome**				
SAPS II	51 (40–62)	49 (38–61)	53.5 (44–62)	0.14
ARDS (PaO2/FiO2 < 200)	80 (77.7)	51 (73.9)	29 (85.3)	0.19
Mechanical ventilation (days)	12 (4–23)	14 (4–23)	8 (2.3–18)	0.16
Ventral decubitus	63 (61.2)	39 (56.5)	24 (70.6)	0.17
ECMO	9 (8.7)	5 (7.3)	4 (11.8)	0.67
Dialysis	23 (22.3)	17 (24.6)	6 (17.7)	0.42
Pulmonary embolism	10 (9.7)	3 (4.4)	7 (20.6)	0.03 *
ICU LOS (days)	16 (5–29)	18 (8–33)	8 (4–18)	0.01 *
In-hospital LOS (days)	25 (11–39)	30 (18–42)	8 (4–19)	<0.01 *

Data are all expressed as median (Q1–Q3) or n/N (%) where N is the total number of patients with available data. * *p* < 0.05. Abbreviations: ID = immunodeficiency, min = minute, L = liter, ARF = acute respiratory failure (clinical), EMS = emergency medical services (medicalized), CRP = C-reactive protein, CT = computed tomography, ICU = intensive care unit, LOS = length of stay, ARDS = acute respiratory distress syndrome, ECMO = extracorporeal membrane oxygenation, SAPS II = Simplified Acute Physiology Score II.

**Table 2 jcm-09-03744-t002:** Characteristics of the study population and comparison between Type 1 and Type 2 clinical profiles.

General Characteristics	TYPE 1 N = 47	TYPE 2 N = 56	*p*-Value
Age (years)	65 (58–75)	70 (62–73)	0.39
Male	39 (83)	42 (75)	0.33
**Comorbidities**			
Hypertension	30 (63.8)	31 (55.4)	0.38
Cardiovascular diseases	14 (29.8)	22(39.3)	0.31
Diabetes	13 (27.7)	21 (37.5)	0.29
Chronic kidney diseases	6 (12.8)	10 (17.9)	0.48
Malignencied or ID	6 (12.8)	6 (10.7)	0.75
Respiratory diseases	13 (27.7)	23 (41.1)	0.16
**Pre-hospital parameters**			
Respiratory rate (/min)	28 (23.5–31)	35 (30–40)	<0.01 *
1st oxygen saturation (%)	79 (70–82.5)	65 (51.5–80)	<0.01 *
Oxygen need (L/min)	15 (9–15)	15 (15–15)	0.03 *
Heart rate (bpm)	90 (80–113)	110 (92–121)	0.02 *
Systolic blood pressure (mmHg)	119 (100–130)	128 (110–140)	0.02 *
Glasgow score scale	15 (15–15)	15 (14–15)	0.04 *
Temperature (°C)	38 (37.5–39)	38 (37–38.6)	0.28
Shock	7 (17.9)	11 (19.6)	0.53
Confusion	6 (12.8)	19 (33.9)	0.01 *
**Timing**			
Since 1st symptom (days)	7 (5–10)	7 (3.8–8.5)	0.56
Since ARF (days)	1 (1–2)	1 (0.8–2)	0.26
**Pre–hospital care**			
Intubation in the field	27 (71.1)	53 (98.2)	<0.01 *
**Laboratory findings**			
Creatinine (µmol/L)	96 (75–136)	97 (77–164)	0.51
CRP (mg/L)	139 (73–222.5)	150 (98–224)	0.64
Lymphocytes (/µL)	780 (520–1060)	600 (430–1015)	0.17
Lactate (mmol/L)	1.5 (1.0–2.1)	1.7 (1.2–2.3)	0.30
**Radiology findings**			
Typical CT-scan	25 (67.6)	15 (46.9)	0.08
Extension >50%	15 (45.5)	23 (79.3)	<0.01 *
**ICU stay and outcome**			
SAPS II	43.5 (36–58)	53 (42–63.3)	0.04 *
ARDS (PaO2/FiO2 < 200)	36 (76.6)	44 (78.6)	0.81
Mechanical ventilation (days)	15 (4.5–25)	9.5 (3.0–19)	0.55
Ventral decubitus	30 (63.8)	33 (58.9)	0.61
ECMO	6 (12.8)	3 (5.4)	0.33
Dialysis	9 (19.2)	14 (25)	0.48
Pulmonary embolism	6 (12.8)	4 (7.1)	0.53
ICU LOS (days)	17 (5.5 – 30.5)	15.5 (4.5 – 25.3)	0.42
**Outcome**			
In-hospital LOS (days)	26 (12–37.5)	20.5 (9–40)	0.36
ICU mortality	15 (31.9)	17 (30.4)	0.86
In-hospital mortality	15 (31.9)	19 (33.9)	0.83

Data are all expressed as median (Q1–Q3) or n/N (%) where N is the total number of patients with available data. Type 1 is defined in our study as patients with “silent/happy” hypoxemia without any symptoms of ARF. Type 2 corresponds to patients who have both hypoxemia and clinical ARF. * *p* < 0.05. Abbreviations: ID = immunodeficiency, min = minute, L = liter, ARF = acute respiratory failure (clinical), EMS = emergency medical services (medicalized), CRP = C-reactive protein, CT = computed tomography, ICU = intensive care unit, LOS = length of stay, ARDS = acute respiratory distress syndrome, ECMO = extracorporeal membrane oxygenation, SAPS II = Simplified Acute Physiology Score II.

**Table 3 jcm-09-03744-t003:** Multivariate analysis of factors associated with the difference between Type 1 and Type 2 clinical profiles.

General Characteristics	Odds Ratio	95% CI	*p*-Value
Age (years)	1.00	0.99–1.01	0.93
Male	1.07	0.75–1.52	0.71
**Comorbidities**			
Hypertension	0.95	0.74–1.22	0.70
Cardiovascular diseases	0.73	0.55–0.96	0.03 *
Diabetes	1.03	0.82–1.29	0.82
Chronic kidney diseases	0.92	0.78–1.09	0.36
Malignencied or ID	0.99	0.70–1.39	0.94
Respiratory diseases	1.08	0.82–1.42	0.59
**Pre-hospital parameters**			
Respiratory rate (/min)	0.98	0.96–1.00	0.02 *
1st oxygen saturation (%)	1.00	0.99–1.01	0.91
Oxygen need (L/min)	0.99	0.96–1.03	0.73
Heart rate (bpm)	1.00	0.99–1.00	0.33
Systolic blood pressure (mmHg)	1.00	1.00–1.01	0.35
Glasgow score scale	1.02	0.97–1.07	0.41
Temperature (°C)	1.05	0.94–1.17	0.42
Shock	0.79	0.49–1.27	0.34
Confusion	0.78	0.57–1.07	0.14
**Timing**			
Since 1st symptom (days)	1.02	0.99–1.04	0.17
Since ARF (days)	0.99	0.93–1.06	0.74
**Pre-hospital care**			
Intubation in the field	0.67	0.51–0.88	<0.01 *
**Laboratory findings**			
Creatinine (µmol/L)	1.00	1.00–1.00	0.94
CRP (mg/L)	1.00	1.00–1.00	0.27
Lymphocytes (/µL)	0.85	0.68–1.07	0.18
Lactate (mmol/L)	1.06	1.02–1.11	0.01 *
**Radiology findings**			
Typical CT-scan	1.55	1.22–1.96	<0.01 *
Extension >50%	0.76	0.61–0.97	0.03 *
**ICU stay**			
SAPS II	1.00	0.99–1.01	0.39
ARDS (PaO2/FiO2 < 200)	0.87	0.62–1.20	0.40
Mechanical ventilation (days)	1.03	1.01–1.06	0.02 *
Ventral decubitus	1.20	0.88–1.63	0.27
ECMO	0.93	0.54–1.59	0.79
Dialysis	1.06	0.79–1.42	0.70
Pulmonary embolism	1.25	0.87–1.79	0.24
ICU LOS	0.95	0.93–0.98	<0.01 *
**Outcome**			
In-hospital LOS	1.02	1.00–1.05	0.04 *
ICU mortality	1.07	0.72–1.59	0.75
In-hospital mortality	1.07	0.72–1.59	0.74

Type 1 is defined in our study as “silent/happy” hypoxemia without symptoms of ARF. Type 2 corresponds to patients who have both hypoxemia and clinical ARF. * *p* < 0.05. Abbreviations: ID = immunodeficiency, min = minute, L = liter, ARF = acute respiratory failure (clinical), EMS = emergency medical services (medicalized), CRP = C-reactive protein, CT = computed tomography, ICU = intensive care unit, LOS = length of stay, ARDS = acute respiratory distress syndrome, ECMO = extracorporeal membrane oxygenation, SAPS II = Simplified Acute Physiology Score II.

## Data Availability

The database of the study will be freely accessible online within 3 months after publication, upon reasonable request to the corresponding author.

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
