# Peer review of "Pre-Hospital Management of Critically Ill Patients with SARS-CoV-2 Infection: A Retrospective Multicenter Study"

_jcm, 2020, doi:10.3390/jcm9113744_

Round 1

Reviewer 1 Report

Major comments

  1. What is the point of comparing the prognosis between phenotype 1 and 2? You mention in the context that the difference between phenotype 1 and 2 is whether the patient has acute respiratory failure or not. So, the clinical and radiological differences between these two do not sound informative.
  2. What is the dependent variable on the multivariate analysis? Odds ratio of what? If you are trying to calculate the risk factors for having phenotype 2, the study population has to be all infected patients rather than just those who were put on ICU through EMS. There might have been many patients who were admitted to hospitals via emergency departments or local clinics with mild cough/fever/etc. (and thus were excluded from this study) and eventually developed acute respiratory failure afterward.
  3. You can not say that ICU and in-hospital mortality was not affected by clinical phenotype (P10L244-5). P-valure > 0.05 does not mean that there is no difference. Rather, it only shows that we do not have data sufficient enough to say that there is a significant difference.

Minor comments

  1. The last paragraph of Data Collection section (P4L114-7) has to be moved to analysis section.
  2. You need to show 95% CI, not just p-value, both on the main text and tables.
  3. You need to clarify the definition for acute respiratory failure (is it the same as ARDS?) and restlessness/confusedness. 
  4. On line 239-41, if there is no difference found, then why do you have OR with the p-values 0.02 and <0.01, both less than 0.05?

Author Response

Reviewer 1

English language and style

(x) Extensive editing of English language and style required
( ) Moderate English changes required
( ) English language and style are fine/minor spell check required
( ) I don't feel qualified to judge about the English language and style

Yes

Can be improved

Must be improved

Not applicable

Does the introduction provide sufficient background and include all relevant references?

( )

(x)

( )

( )

Is the research design appropriate?

( )

(x)

( )

( )

Are the methods adequately described?

( )

( )

(x)

( )

Are the results clearly presented?

( )

( )

(x)

( )

Are the conclusions supported by the results?

( )

(x)

( )

( )

Comments and Suggestions for Authors

Major comments

  1. What is the point of comparing the prognosis between phenotype 1 and 2? You mention in the context that the difference between phenotype 1 and 2 is whether the patient has acute respiratory failure or not. So, the clinical and radiological differences between these two do not sound informative.

We express our respectful appreciation towards the first reviewer’s comments. By comparing the two phenotypes (with or without acute respiratory failure), our intention was to identify a difference in patient outcome or mortality depending on whether ARF was present in the initial pre-hospital phase. All emergency and critical medicine physicians during the ongoing pandemic oversaw and treated COVID-19 patients with altered respiratory parameters (low saturation, low pO2) and without any visible clinical respiratory distress signs, yet they required an increase of oxygen / ventilation therapies. This presentation was constant during acute pre-hospital phases, upon arrival at the emergency room and/or at admission to the ICU. Thus, through the prism of pre-hospital care, we intended to differentiate these two sub-populations of patients (both hypoxemic, requiring high oxygen support and frequently admitted to the ICU). Gattinoni et al (ICM, 2020) described two distinct phenotypes (L and H); they showed interest in ventilation parameters of ARDS and imaging findings, yet the initial clinical presentation and management of these patients remain to be explored. This is why we chose to call these presentations “phenotypes 1 and 2” (hypoxemia with or without ARF). Attending to these "happy hopoxemic" patients is a subject that was regularly questioned in the field. The experience of the first wave and these data (despite the limited number of patients) encourage us to care for patients presenting hypoxemia without clinical respiratory distress in the same way we care for patients presenting ARF. However, it is also necessary to take into account recent literature regarding this subject, especially regarding alternatives to invasive mechanical ventilation (NIV and high flow oxygen therapy). The clinical and radiological findings we provided appear valuable, to us, in this context since it invokes a probable continuum of the disease (radiological damage appears to be more severe when hypoxemia and ARF are present).

  1. What is the dependent variable on the multivariate analysis? Odds ratio of what? If you are trying to calculate the risk factors for having phenotype 2, the study population has to be all infected patients rather than just those who were put on ICU through EMS. There might have been many patients who were admitted to hospitals via emergency departments or local clinics with mild cough/fever/etc. (and thus were excluded from this study) and eventually developed acute respiratory failure afterward.

Clearly we did not claim the ability to predict different phenotypes with our study findings, rather we intended to highlight the relevant phenotype / clinical presentation criteria that can guide emergency and critical medicine physicians attending to the most severely ill patients. This is why we have decided to analyze the patients by the management system, i.e. pre-hospital management.

You can not say that ICU and in-hospital mortality was not affected by clinical phenotype (P10L244-5). P-value > 0.05 does not mean that there is no difference. Rather, it only shows that we do not have data sufficient enough to say that there is a significant difference.

Indeed, we have weighted our sayings (P10, L244-5), the size of this study is relatively small, yet it is one of the only pre-hospital cohorts available in literature. Of course, these results need to be confirmed by research on a larger sample size. And actually, a larger study on this subject is underway on patients admitted to the emergency department without prior pre-hospital medicalized care.

Minor comments

  1. The last paragraph of Data Collection section (P4L114-7) has to be moved to analysis section.

The paragraph mentioned was moved to Analysis section as requested

  1. You need to show 95% CI, not just p-value, both on the main text and tables.

We presented 95% CI along with p-value in all our results (mainly in the manuscript), as requested.

  1. You need to clarify the definition for acute respiratory failure (is it the same as ARDS?) and restlessness/confusedness. 

We are essentially talking about respiratory distress or acute respiratory failure, which is only based on clinical signs (no biological or radiological criteria). We are going to specify this in our paper by adding the following to the inclusion criteria: "the patients included should present clinical respiratory distress associating polypnoea, signs of respiratory effort, use of accessory respiratory muscles, difficulty in speaking, etc." ARDS syndrome is just used at the end in the outcome results in order to describe the evolution of the study population.

  1. On line 239-41, if there is no difference found, then why do you have OR with the p-values 0.02 and <0.01, both less than 0.05?

The data cited and provided in this paragraph are significant, we have added words such as “except” or “outside of” to clarify that the only significant findings are what we provided in the text.

We also did an extensive editing of English language and style of the revised manuscript

Best regards

Reviewer 2 Report

Dear Authors

Greetings

Your research is very interesting, congrats. So I would like to suggest a better presentation for the table of results. I agree that tables are generally best if you want to be able to look up specific information or if the values must be reported precisely. However graphics are best for illustrating trends and making comparisons. I would like to suggest to improve the discussion comparing with some previous studies about the sequence diversity in SARS-CoV-2 proteins associated with the virus pathogenicity or transmission in Europe (several countries would be associated with specific or several virus clades or mutations and that the frequency in clinical symptoms of individuals with COVID-19 also varies between these countries). I suggest to compare too with other studies that include patient clinical information (sequence polymorphisms and symptoms) suggesting that the sequence determination of the associated infectious genome should be combined to give better understanding of SARS-CoV-2 pathogenicity and help in the development of adapted treatments. Maybe that would be a good way for some hypothesis your study inspire us. 

Best regards

Author Response

Reviewer 2

Open Review

English language and style

( ) Extensive editing of English language and style required
( ) Moderate English changes required
(x) English language and style are fine/minor spell check required
( ) I don't feel qualified to judge about the English language and style

Yes

Can be improved

Must be improved

Not applicable

Does the introduction provide sufficient background and include all relevant references?

(x)

( )

( )

( )

Is the research design appropriate?

(x)

( )

( )

( )

Are the methods adequately described?

(x)

( )

( )

( )

Are the results clearly presented?

(x)

( )

( )

( )

Are the conclusions supported by the results?

(x)

( )

( )

( )

Comments and Suggestions for Authors

Dear Authors

Greetings

Your research is very interesting, congrats. So I would like to suggest a better presentation for the table of results. I agree that tables are generally best if you want to be able to look up specific information or if the values must be reported precisely. However graphics are best for illustrating trends and making comparisons.

Dear reviewer 2

We are grateful for your comments and suggestions, thank you.

First off, we added a graph to the manuscript, trusting your opinion and leaving it up to the reviewers to decide how best to present the results (figue 1 instead of table 3 added to the manuscript).

Figure 1. Multivariate analysis of factors associated with the difference between type 1 and type 2 clinical profiles.

Type 1 is defined in our study as “silent/happy” hypoxemia without symptom of ARF. Type 2 corresponds to patients who have both hypoxemia and clinical ARF. * p<0.05. Abbreviations: ID= immunodeficiency, min= minute, L= liter, blood P= blood pressure, ARF= acute respiratory failure (clinical), EMS= emergency medical services (medicalized), CRP= C-reactive protein, CT= computed tomography, ICU: intensive care unit, LOS: length of stay, ARDS= acute respiratory distress syndrome, ECMO= extracorporeal membrane oxygenation, SAPS II: Simplified Acute Physiology.

I would like to suggest to improve the discussion comparing with some previous studies about the sequence diversity in SARS-CoV-2 proteins associated with the virus pathogenicity or transmission in Europe (several countries would be associated with specific or several virus clades or mutations and that the frequency in clinical symptoms of individuals with COVID-19 also varies between these countries). I suggest to compare too with other studies that include patient clinical information (sequence polymorphisms and symptoms) suggesting that the sequence determination of the associated infectious genome should be combined to give better understanding of SARS-CoV-2 pathogenicity and help in the development of adapted treatments. Maybe that would be a good way for some hypothesis your study inspire us. 

Secondly, in order to improve the Discussion, we added the following paragraph as well as 2 references:

There may also have been different phenotypes related to the sequence diversity in SARS-CoV-2 proteins associated with viral pathogenicity and transmission in Europe (several countries could be associated with a certain, or even several, virus clades and mutations; frequency of COVID-19 symptoms also varies between these countries) [29]. In addition, one of the emerging therapeutic approaches could be guided by other studies including patient clinical data (sequence polymorphisms and symptoms). Thus, suggesting that sequence determination of the associated infectious genome should be combined with clinical data to give a better understanding of SARS-CoV-2 pathogenicity [30].

Best regards

Reviewer 3 Report

In the manuscript by Le Borgne et al., the authors compared the characteristics and therapeutic management between COVID-19 patients with two different ARDS phenotypes. The authors found distinguishable clinical presentations in these patients before hospitalization, but no difference in mortality among these two phenotypes. The finding suggests an identical management of the patients regardless of clinical phenotypes. The study is well designed, the data presented clearly, and the discussion is comprehensive. The findings in this study could be beneficial to the field. There are only minor comments.

Abbreviation needs to be spelled out in the Abstract, such as EMS and ARDS.

On line 148, the values “60-74” representing the range or interquartile range should be mentioned in the text. And it’s also for those on lines 165, 179 – 181, and 194.

Author Response

Reviewer 3

Open Review

English language and style

( ) Extensive editing of English language and style required
( ) Moderate English changes required
(x) English language and style are fine/minor spell check required
( ) I don't feel qualified to judge about the English language and style

Yes

Can be improved

Must be improved

Not applicable

Does the introduction provide sufficient background and include all relevant references?

(x)

( )

( )

( )

Is the research design appropriate?

(x)

( )

( )

( )

Are the methods adequately described?

(x)

( )

( )

( )

Are the results clearly presented?

(x)

( )

( )

( )

Are the conclusions supported by the results?

(x)

( )

( )

( )

Comments and Suggestions for Authors

In the manuscript by Le Borgne et al., the authors compared the characteristics and therapeutic management between COVID-19 patients with two different ARDS phenotypes. The authors found distinguishable clinical presentations in these patients before hospitalization, but no difference in mortality among these two phenotypes. The finding suggests an identical management of the patients regardless of clinical phenotypes. The study is well designed, the data presented clearly, and the discussion is comprehensive. The findings in this study could be beneficial to the field. There are only minor comments.

Dear reviewer 3, we are humbled by your kind review. During this looming second pandemic wave, we, indeed, believe that our findings will be useful for the daily management of acute patients in pre-hospital phases, in the emergency departments and ICUs.

Abbreviation needs to be spelled out in the Abstract, such as EMS and ARDS.

We have corrected that issue, all abbreviations are spelled out in the newly revised manuscript.

On line 148, the values “60-74” representing the range or interquartile range should be mentioned in the text. And it’s also for those on lines 165, 179 – 181, and 194.

We corrected the text accordingly, these are, indeed, median and quartiles 1 and 3 (Q1-Q3) values. Best regards.

Round 2

Reviewer 1 Report

Thank you for addressing all comments in the review. There seems to be no more corrections/revisions that have to be made.